# The Effects of Nutrition and Health Claim Information on Consumers’ Sensory Preferences and Willingness to Pay

**DOI:** 10.3390/foods11213460

**Published:** 2022-11-01

**Authors:** Xinyi Hong, Chenguang Li, Liming Wang, Zhifeng Gao, Mansi Wang, Haikuan Zhang, Frank J. Monahan

**Affiliations:** 1School of Innovation and Entrepreneurship, Entrepreneurship Institute, Guangzhou University, Guangzhou 510006, China; 2School of Agriculture and Food Science, University College Dublin, D04V1W8 Dublin, Ireland; 3School of Economics and Management, Beijing University of Technology, Beijing 100124, China; 4Irish Institute for Chinese Studies, University College Dublin, D04V1W8 Dublin, Ireland; 5Department Food and Resource Economics, University of Florida, Gainesville, FL 32611, USA; 6School of Economics and Management, East China University of Technology, Nanchang 330000, China

**Keywords:** nutrition claim, health claim, sensory test, choice-based conjoint experiment, payment estimation

## Abstract

As marketing tools, nutrition claims (NCs) and health claims (HCs) can be used to convey the nutritional properties and health benefits of food to consumers, but their respective effects on consumers’ perceptions of healthier meat products are inconsistent in the literature. Using a physical prototype of omega-3-enriched sausages as a research interest, this paper explores how HCs and NCs differently influence consumers’ sensory preferences and willingness to pay (WTP). Sensory tests were carried out among 330 participants, followed by a choice-based conjoint (CBC) experiment to measure consumers’ WTP. Results indicate that, in comparison with the uninformed condition, labeling an omega-3 nutrition claim increased consumers’ sensory liking for omega-3-enriched sausages in the attributes of appearance and texture. Moreover, consumers were willing to pay more for healthier sausages, but labeling HCs did not significantly improve participants’ WTP for omega-3-enriched sausages more than NCs. Hence, HCs did not significantly outperform NCs, when it comes to positively influencing consumers’ sensory liking and paying intentions for omega-3-enriched sausages. The findings of this study have implications for the meat industry in developing healthier sausage formulations with greater likelihood of success in the market.

## 1. Introduction

Meat and its derivatives, with essential nutritional components, constitute an integral part of the human diet. Recent research efforts have been directed towards developing healthier meat products containing compounds with known health benefits or reduced levels of ingredients with negative implications for consumer health [1,2,3,4]. These products meet consumers’ expectations for more healthiness in meat products and provide potentials market opportunities for the meat industry [5,6,7].

In Europe, meat manufacturers can use nutrition claims (NCs) and/or health claims (HCs) labeling on the front of the food pack to communicate the nutritional properties and health benefits of reformulated meat products to consumers. Claims can be a marketing tool to differentiate these nutritionally improved meat products from their conventional counterparts [8,9,10,11]. Regulation EC No. 1924/2006 (available online: https://eur-lex.europa.eu/legal-content/en/ALL/?uri=CELEX%3A32006R1924, accessed on 1 May 2019) defines NCs as *“any claim which states, suggests or implies that a food has particular beneficial nutritional properties (such as ‘contain’, ‘source of’, ‘free of’ ‘reduced’ and ‘increased’)”*, and HCs as “*any claim that states, suggests or implies that a relationship exists between a food category, a food or one of its constituents and health*”. The difference between NCs and the corresponding HCs is that HCs explicitly indicate an association between a specific food substance to a health-related outcome, and this cause-and-effect relationship is scientifically verified [12,13]. For sales purposes, this distinction could be important, as Ares et al. [14] emphasize that failure to establish an association between the nutrient and its benefit is the main barrier to consumer willingness to try healthier food.

For meat manufacturers, whether HCs can give healthier meat products a better competitive advantage than NCs is still unclear. Some studies indicate that HCs resonate better with consumers than NCs, where products labeled with HCs have higher perceived healthiness, attractiveness, consumption intention [14,15,16], and willingness to pay (WTP) [17,18] associated with them. For instance, WTP for frankfurter sausages is less for a low-fat nutrition claim than its corresponding health claim, which relates to reducing cardiovascular disease risks [17]. Similarly, Kallas, Realini, and Gil [16] found that awareness of the health benefits of omega-3 fatty acids positively impacts consumer acceptance of omega-3-enriched beef steak more than a nutrition claim scenario. Nevertheless, some other studies argue that HCs are not superior to NCs in making healthier meat more appealing to consumers. In some cases, NCs outperform HCs in terms of consumers’ purchasing and paying intentions, such as in lean beef steak [9], fortified yogurts [19], orange juice, and milk chocolate [20]. Furthermore, a few studies conclude that NCs and HCs impose negative influences on the perceived healthiness of food due to consumers’ suspicion and unfamiliarity [21]. Hence, given the lack of consistency in the conclusions, more research on the topic of consumers’ preferences for meat products with NCs or HCs is required.

In addition, although Kaur, Scarborough, and Rayner [13] conclude that consumers are more likely to buy and pay for food carrying NCs and/or HCs than conventional counterparts, previous studies find positive, zero, or even negative price premiums in terms of meat products with NCs and/or HCs [9,17,22,23]. Therefore, to achieve economic returns from meat innovation, Decker and Park [24] raise the question of whether consumers are willing to pay more, or if meat companies have to bear the cost of producing healthier meat products.

Numerous studies have stated that maintaining sensory pleasantness and reasonable pricing are two critical factors in consumer acceptance of healthier meat products [24,25,26,27,28,29,30]. Furthermore, evidence suggests that consumers’ sensory perceptions could be altered by information about NCs and/or HCs, such as claims of “plant sterol-enriched” on deli-style turkey slices [22] and “salt-reduced” on cooked ham [31].

Therefore, this study assessed consumers’ perceptions of NCs and HCs made on healthier meat products, using omega-3-enriched sausages as a research interest. Omega-3 fatty acids are nutrients proven to be beneficial for human health and contribute to the normal function of the heart and brain and the maintenance of normal vision. These functions are recognized by the European Food Safety Authority (EFSA) (regulation No. 432/2012) as permitted health benefits. In addition, several scientific papers report that sausages are technologically appropriate for omega-3 enrichment [32,33]. The main objectives of this paper are two-fold: first, to assess whether and how different omega-3 information scenarios of NCs and HCs influence sensory preferences and WTP for omega-3-enriched sausages; second, to estimate how much a consumer is willing to pay for healthier sausages. To achieve the first objective, a within-subject comparison between two rounds of sensory tests was implemented to explore how sensory preferences are changed by different claim information. Starting with a blind sensory test, participants were given omega-3-enriched sausages with no information. This was followed by an informed test, in which participants were informed of the omega-3 enrichment in the sausages with either a nutrition claim (further referred to as N group) or a health claim (further referred to as H group). To assess the second objective, a choice-based conjoint (CBC) experiment was carried out immediately after the two sensory tests, and consumers’ WTP for four types of healthier sausages (i.e., higher meat content, source of omega 3, reduced fat, reduced salt) was estimated.

This study aims to find out if NCs and HCs can generate better sensory experiences and elicit higher WTP. The findings from this study are expected to deepen the understanding of consumers’ perceptions of NCs and HCs, and provide insights for meat manufacturers to develop and market healthier meat products with a greater chance of success.

## 2. Materials and Methods

### 2.1. Data Collection

This study was approved by the Human Research Ethics Committee for Sciences at University College Dublin (UCD) (LS-17-91-Hong-Li). The products tested were safe for consumption. The preparation of omega-3 sausage samples is described in Appendix A. The data collection was carried out in an ISO-standard sensory laboratory designed in accordance with ISO 8589:2007 at the UCD Institute of Food and Health. There were five phases to be completed in one session: (i) a blind sensory test of omega-3-enriched sausages; (ii) establishing participants’ familiarity with omega-3 fatty acids and randomly informing participants of an omega-3 claim (either a nutrition claim or a health claim); (iii) an informed sensory test of omega-3-enriched sausages; (iv) a CBC experiment requiring participants to choose their preferred sausage options; (v) collecting information regarding sociodemographic backgrounds and consumption habits of processed meat and dietary supplements. Figure 1 depicts the experimental procedure. Each step is described in greater detail throughout this section.

Participants were recruited via email and advertisement on the UCD campus on a voluntary basis, and all were untrained consumers of processed meat and over 18 years of age. Three-hundred and thirty participants (326 valid, 141 men and 185 women) were involved, and they all gave informed consent via the statement “I am aware that my responses are confidential, and I agree to participate in this survey”. They could withdraw from the survey at any time without giving a reason. The tests undertaken are specified in Section 2.2 and Section 2.3. More details on participants’ profiles and characteristics are presented in Section 3.1.

### 2.2. Sensory Tests

Both the blind and informed sensory tests of omega-3 sausages were conducted in standard sensory booths under artificial daylight-type illumination and a temperature of 22–24 °C with an air circulation system.

The same participants did both the blind and informed tests. For all participants, the blind test was conducted first. Under the blind test, participants were given omega-3-enriched sausage samples labeled with three-digit random numbers (three-digit numbers did not have a zero-starting point), but they were not given any additional information on sausage samples. Samples were served in sequential monadic order [34]. A raw sausage sample and a cooked sausage sample were presented to participants in pairs on a food tray, where the raw was sealed in a transparent bag for observation to mimic the situation on the shop shelf, and the cooked was served on odorless white plastic plates. Participants were asked to observe the raw sausage and to taste the cooked sausage. Sensory liking on the characteristics of raw sausage appearance and sausage texture, taste, and overall sensory liking were rated, using a Likert 9-point scale [35,36], where 1 = “dislike extremely”, 2 = “dislike very much”, 3 = “dislike moderately”, 4 = “dislike slightly”, 5 = “neither like nor dislike”, 6 = “like slightly”, 7 = “like moderately”, 8 = “like very much”, and 9 = “like extremely”. RedJade Sensory Software (RedJade Sensory Software Official Website: https://redjade.net/, accessed on 1 May 2019), which was installed on the computers in each booth, was employed to collect responses. Still mineral water and plain crackers were available for palate cleansing between sausage samples and after tasting.

After participants completed the blind test and before the informed test started, participants’ prior knowledge of omega-3 was self-assessed on a Likert 9-point scale [35,36], where 1 = ‘‘extremely unfamiliar”, 5 = “neither unfamiliar nor familiar”, and 9 = ‘‘extremely familiar”. This was followed by the information disclosure of one of two claims. The information on an omega-3 nutrition claim was stated as follows:

“Please note that the sausages you will receive: (1) have sufficient **omega-3 fatty acids** to bear a European Food Safety Authority (EFSA) nutrition claim; (2) can have “**source of omega-3 fatty acids**” on their package”.

The information on an omega-3 health claim (in the nutrition and health claims, the wording is quoted from Regulation EU No. 432/2012) was stated as follows:

“Please note that the sausages you will receive: (1) have sufficient **omega-3 fatty acids** to bear a European Food Safety Authority (EFSA) nutrition claim and health claim; (2) can have “**source of omega-3 fatty acids**” and “**omega-3 fatty acids contribute to the normal function of the heart**” on their package”.

Participants were randomly informed of either a nutrition claim (N group) or a health claim (H group). Following this disclosure of information, participants were given a raw and a cooked omega-3-enriched sausage sample with the same ingredients as above. Again, participants were asked to rate their sensory liking of the raw sausage appearance and the cooked sausage texture, taste, and overall sensory liking on a Likert 9-point scale.

### 2.3. Choice-Based Conjoint (CBC) Experiment

After the informed test, a CBC experiment was used to measure consumer preferences and WTP for the selected attributes of sausages. The CBC experiment is a widely applied method to investigate consumers’ preferences towards food claims and other attributes of healthier meat products in the existing literature [23,37,38]. In this study, five attributes of sausage products (i.e., price, meat content, omega-3 enrichment, fat reduction, and salt reduction) and their associated levels are described in Table 1.

These attributes and levels were selected to reflect sausage characteristics valued by consumers, based on the relevant literature [38,39,40,41]. Price and meat content ranges were drawn on real commercial products in Irish grocery shops or supermarkets (see Appendix B: A summary of selected information on pork sausages available in Irish supermarkets). The three nutrition claims were selected to measure consumers’ WTP for three types of healthier sausages, where “reduced fat” and “reduced salt” were two popular reformulation approaches regarding healthier processed meat products. Moreover, the “source of omega-3” was meant to explore how different health-related claims impact WTP for omega-3-enriched sausages.

A full-factorial set of attribute-level combinations produced 128 product alternatives (4*4*2*2*2). Considering cognitive burdens, a modified Fedorov algorithm [42] was used to reduce the number to 24 alternatives, which was further grouped into 12 choice sets in the questionnaire (see Appendix C). A “neither” option was also added in each choice set, which allowed a participant not to “purchase” sausages. Participants were asked to select the most desirable alternative in the choice set as if it were a real shopping experience and repeated their decision-making process for all 12 choice sets. 

### 2.4. Econometric Models

Choices made by participants in the conjoint experiment manifest the utilities of each choice, which quantitatively measures consumers’ preferences on listed attributes, according to Lancaster consumer theory and random utility theory [43,44]. A mathematical denotation of utility is modeled as below:(1)$$U_{njt}=V_{njt}\;+\varepsilon_{njt}\;$$
where $U_{njt}$ denotes the total utility obtained by a consumer *n*, from the alternative *j* (*j* = 1, ..., J) in the choice set *t*. $V_{njt}$ measures utility by a vector of explainable variables constructed by researchers. $\varepsilon_{njt}$ represents the difference between the measured utility and the total utility.

Under the utility-maximization assumption, a consumer *n* chooses an alternative *i* among all J alternatives (*j* = 1, ..., J) within the same choice set *t*, if and only if $U_{nit}\;$> $U_{njt}$ ∀ j≠i for any *i* and *j*. Derivations from random utility models lead to probabilistic choice behavior. The probability of alternative *i* being chosen is framed as:(2)$$P_{nit}=\;\mathrm{Prob}\;(U_{nit}\;>\;U_{njt}\forall \;j\neq i)\;=\mathrm{Prob}\;(V_{nit}\;+\varepsilon_{nit}\;>V_{njt}+\varepsilon_{njt}\;\forall \;j\neq i)\;=\;\mathrm{Prob}\;(\varepsilon_{njt}\;-\varepsilon_{nit}\;<\;V_{nit}\;-\;V_{njt}\;\forall \;j\neq i)\;$$

Researchers’ identification of the distribution of the unobserved utility component $\;\varepsilon_{njt}$ and specification of its density function decides the form of cumulative probability function in Equation (2) to analyze choice experiment data. As previous studies suggest that consumers have heterogeneous preferences for meat content in sausages and health-related food claims are heterogeneous according to existing literature [9,21,38,45,46], coefficients of observed variables should be allowed to vary among participants. As a result, a generalized, highly flexible, random-parameter logit (RPL) model is employed for data analysis. Under an RPL model, the unobserved portion of utility is decomposed into two parts: a part taking a closed form plus a part that is subject to any specified distribution, which not only relaxes limitations on taste variation, substitution patterns, and panel data, but also allows the form of distribution settings in the parameters suitable for research requirements [47]. Each consumer is treated with a set of specific parameters reflecting individual preference and variance of random parameters leads to correlation in unobserved utility across alternatives [48]. The standard specification of $V_{njt}$ is constructed linearly with parameters [16]. Under RPL modeling, Equation (1) is illustrated as:(3)$$U_{njt}={\beta^{\prime }}_{n}X_{njt}+\varepsilon_{njt}\;$$
where ${\beta^{\prime }}_{n}$ is a vector of parameter coefficients associated with a participant, representing the decision maker’s personal tastes [44]. However, the values of ${\beta^{\prime }}_{n}$ are not known. Therefore, with the assumption that $\varepsilon_{njt}$ follows type-I extreme value distribution, the unconditional choice probability function is derived as the integral of a standard logit model weighted by a density function of parameters.
(4)$$P_{nit}=\int \left(\frac{\exp\left(\beta^{\prime }X_{nit}\right)}{\sum_{j}\exp\left(\beta^{\prime }X_{njt}\right)}\right)f(\beta |\theta )d\beta$$
where $P_{nit}$ denotes the choice probability of consumer *n* choosing alternative *i* in the choice set *t*. f(β|θ) is a density function of parameters θ in the constructed model. Without a closed form, the integral needs to be estimated by simulation. Simulated probabilities of a sequence of choices are performed in the log-likelihood framework, where values of parameters are taken from numerous times of Halton draws [49].

In this study, the utility function of a consumer *n*, selecting the sausage product *j* (*j =* 1, 2, and 3 for Product 1, Product 2, and neither) in the choice set *t* (*t =* 1, 2, 3, …, 12), is expressed as:(5)$$U_{njt}=\beta_{0}+\beta_{1}Price_{jt}+\beta_{2}Meat_{jt}+\beta_{3}Om3_{jt}+\beta_{4}Rfat_{jt}+\beta_{5}Rsalt_{jt}+\varepsilon_{njt}$$

Dependent variables are dichotomous, where 1 equals the alternative being chosen and 0 otherwise. The constant $\beta_{0}$ captures the effect of the opt-out option and represents the utility level if a consumer chooses the “neither” (i.e., not purchasing) option. Both the price variable and the meat content variable have four levels (see Table 1). Nutrition claims regarding omega-3, reduced fat, and reduced salt (corresponding to $Om3_{jt}$, $Rfat_{jt}$, and $Rsalt_{jt}$ in Equation (5), respectively) are all binary variables, where 1 indicates the sausage alternative includes the respective nutrition claim and 0 otherwise. Only main effects are evaluated in the models.

The price coefficient is used to calculate WTP for the remaining sausage attributes [50]. Compared with estimation in preference space, estimation in WTP space is advantageous in reflecting more realistic WTP measurement and better goodness of fit in data [51,52,53,54]. Equation (5) is reparameterized as:(6)$$U_{njt}=\beta_{n0}/\mu_{n}+\beta_{n1}/\mu_{n}Price_{njt}+\sum_{m=2}^{5}\left(\beta_{nm}/\mu_{n}\right)X_{njt}+\varepsilon_{njt}\;$$
(7)$$U_{njt}=\beta_{n0}/\mu_{n}+\lambda_{n}[Price_{njt}+{\mathrm{WTP}}_{n}X_{njt}]+\varepsilon_{njt}\;$$
where $\mu_{n}$ is a scale parameter.$\;\beta_{nm}$ denotes the coefficient vector of nonprice attribute $X_{njt}$ (i.e., $Meat_{njt},\;Om3_{njt},\;Rfat_{njt},\;Rsalt_{njt}$). $\;\varepsilon_{njt}$ is the type-I extreme value distributed error term with scaled variance to be constant (i.e., $\pi^{2}/6$) [55]. The utility coefficients are defined as $\lambda_{n}=\beta_{n1}/\mu_{n},\;c_{n}=\beta_{nm}/\mu_{n}$, and ${\mathrm{WTP}}_{n}=c_{n}/\lambda_{n}.$ The coefficient of price in the utility function is estimated as a lognormal parameter, which is suitable when a higher price is always negatively valued by a consumer [28,44,49,56]. The WTP coefficients of nonprice attributes, including meat content and claims, are estimated as independently random parameters with normal distribution.

### 2.5. Data Analysis

Stata 15.1 (StataCorp. 2017. Stata Statistical Software: Release 15. College Station, TX, USA: StataCorp LLC) software was used to run all data analysis.

Sensory likings were analyzed by paired-sample t-tests to compare the within-subject difference of the two omega-3 sausages’ sensory liking means before vs. after the omega-3 information disclosure.

WTP estimation for healthier sausages (i.e., higher meat content, source of omega 3, reduced fat, reduced salt) was conducted by maximum simulated likelihood [44,49,51], with 2000 Halton draws being performed to ensure robust analytical outputs. Consumers’ WTP for omega-3-enriched sausages in the N group and the H group was estimated to reflect consumers’ WTP with *vs* without the expected health benefit information of omega-3.

## 3. Results and Discussion

### 3.1. Participants

Of the 330 participants who were involved in this study, there were 4 incomplete questionnaires. The remaining 326 valid participants were randomly divided into 2 subgroups based on the two different omega-3 claim information disclosures, and each group had 163 participants. Table 2 reports characteristics of all participants, and the two subgroups (i.e., N group and H group) regarding socio-demographics, omega-3 familiarity, and processed meat consumption habits.

Participants were roughly balanced for gender (56.75% female and 43.25% male) but were unbalanced for age and education. They were biased toward young and highly educated subjects, with most participants being students, which is quite common when volunteer-based food-related consumer studies are conducted on a university campus [22,56]. The average score for self-assessed familiarity with omega-3 was 6, showing that participants were slightly familiar with omega-3. Regarding consumption habits, 79.44% of participants ate processed meat at least once a week. Furthermore, the two subgroups of the N group and the H group consistently had similar characteristics to the total participants. No statistically significant differences (*p* > 0.05) were observed between the N group and the H group regarding socioeconomic and demographic characteristics according to the results of the independent two-sample t-tests. Hence, participants in the N group were comparable with their counterparts in the H group.

### 3.2. Sensory Liking Results

Sensory liking for three sensory attributes (appearance, taste, and texture) and overall sensory liking ratings are detailed in Table 3.

The average liking ratings of appearance, taste, texture, and overall liking were all between 5 (5 = “neither like nor dislike”) and 6 (6 = “like slightly”), showing that participants liked omega-3-enriched sausages but to a certain degree. Although this finding indicates that the omega-3-enriched sausage prototype in this study needs further sensory optimization, consumers’ preferential attitudes towards omega-3-enriched sausages and other omega-3-enriched processed meat products are in line with many previous studies [32,33,40,41]; thus, omega-3-enriched sausages are worth being considered as a product upgrade with market potential.

A within-subject comparison of sensory liking means between the blind test and the informed test indicated that, although the omega-3 information disclosure increased the averaged liking ratings of all three sensory attributes and overall liking, only appearance (*p* = 0.007) and texture (*p* = 0.019) differences were statistically significant among participants in the N group. For participants in the H group, the averaged liking ratings for omega-3 sausages displayed no statistical difference in any sensory aspect after the omega-3 information disclosure compared to before. Hence, the positive impacts on sensory aspects from NCs and HCs were more evident in the case of NCs than HCs. Participants in the N group were significantly more accepting of omega-3 sausages’ appearance and texture after knowing the sausages labeled a “source of omega-3” nutrition claim, but their counterparts in the H group indicated insignificant differences in the acceptability of omega-3 sausages before and after the health claim information disclosure. These findings are somewhat similar to the conclusion of Lahteenmaki, Lampila, Grunert, Boztug, Ueland, Astrom, and Martinsdottir [21] of a negative halo effect from HCs. Lahteenmaki, Lampila, Grunert, Boztug, Ueland, Astrom, and Martinsdottir [21] found that the omega-3 nutrient, per se without a benefit, increased perceived healthiness of products among consumers, but providing them with the expected health benefits information of memory-related or weight-related functions from the omega-3 health claims did not increase the perceived healthiness compared with the conventional products. The authors explained that this negative halo effect from HCs could be due to consumers’ suspicions, uncertainties, and distrust of claimed health rewards. Some more explanations of the negative halo effect from HCs might be due to difficulty in consumers’ understanding of the information in a health claim [45,57] and substitutional effects arising from excessive information presented in dual health and nutrition claims [18]. Although the negative halo effect from HCs was not observed in this study, labeling NCs is better than HCs, in terms of improving participants’ sensory liking for omega-3-enriched sausages in comparison with the uninformed situation, especially in the sensory attributes of appearance and texture liking.

A between-subject comparison of sensory liking means between the N group and the H group under the same information condition gave similar results, where participants in the H group only liked the appearance of omega-3 sausages more than their counterparts in the N group whether they were unaware or aware of the omega-3 enrichment in the sausages. However, in terms of the other two sensory attributes (i.e., taste, and texture) and overall liking, having or not having the expected health benefit information of omega-3 had no statistically significant impacts on participants’ sensory liking. In contrast to some other studies on healthier meat products, our findings suggest that different knowledge information from NCs and HCs should not significantly affect consumers’ sensory liking assessment. However, Grasso, Monahan, Hutchings, and Brunton [22] found that a health information disclosure significantly increased the appearance liking for plant sterol-enriched turkey compared to the blind condition. In addition, Kallas, Realini, and Gil [16] concluded that revealing the health benefits of omega-3 mitigates some negative impacts of beef visual appearance defects. Therefore, whether HCs information can generate some more positive impacts on sensory liking than NCs, especially for the attribute of the appearance of healthier meat products, could be product specific.

### 3.3. WTP Estimation

Table 4 reports the WTP estimation of the selected attributes by the two information groups (i.e., N group and H group) based on the choice experiments.

Overall, participants were willing to pay significantly more for all healthier sausages included in this study. In the N group, participants were willing to pay most for a “reduced fat” claim (an additional EUR 0.59 per 454 g), followed by a “source of omega-3” claim (EUR 0.50) and a “reduced salt” claim (EUR 0.39) and the least for a higher meat content (EUR 0.32). In the H group, participants indicated the highest WTP for a “source of omega-3” (EUR 0.59) claim, followed by a “reduced fat” (EUR 0.43) and a “reduced salt” claim (EUR 0.43) and the least for a higher meat content (EUR 0.31).

These findings are consistent with previous studies’ conclusions that consumers are willing to pay a higher price for healthier sausages, including fiber-added sausages [58], salt-reduced sausages, and fat-reduced sausages [23,59].

Nevertheless, the WTP mean in the H group was marginally higher (EUR 0.09) than in the N group. Although Ares et al. [60] report that lack of nutritional knowledge represents a critical barrier to the acceptability of healthier food and emphasize the importance of using health claims as labels, this study suggested that labeling an omega-3 health claim to convey the knowledge of health benefit of omega-3 to the normal function of the heart should not significantly improve participants’ WTP for omega-3-enriched sausages compared to an omega-3 nutrition claim. One possible explanation could be that participants had some familiarity with omega-3. The average familiarity with omega-3 rating was 6 (Table 2), indicating that for participants omega-3 was not unknown to them. With nutritional knowledge on omega-3, the influence of the health benefits of omega-3 from the health claim was therefore insignificant. Somewhat similarly, Lahteenmaki, Lampila, Grunert, Boztug, Ueland, Astrom, and Martinsdottir [21] compared consumers’ perceptions of the more commonly marketable nutrient “omega-3 fatty acids” with a less marketable nutrient “bioactive peptides”. Their results suggested that informing consumers of HCs only significantly increases the perceived healthiness of food with bioactive peptides, but not with omega-3. Therefore, consumers are willing to pay higher prices for meat companies to produce healthier processed meat products, such as omega-3-enriched, fat-reduced, and salt-reduced sausages. However, in the case of omega-3-enriched sausages, labeling an omega-3 health claim does not significantly outperform an omega-3 nutrition claim, in terms of consumers’ sensory liking and WTP.

## 4. Conclusions

NCs and HCs are important marketing tools to convey nutritional properties and health benefits to consumers. This study provides insights into how NCs and HCs influence consumers’ perceptions of healthier sausages from the perspectives of sensory liking and WTP. Results indicate that consumers are willing to pay a higher price for healthier sausages, but in the case of omega-3-enriched sausages, labeling an omega-3 health claim does not significantly outperform an omega-3 nutrition claim in terms of positively influencing consumers’ sensory liking and WTP.

Therefore, the implication for the meat industry is that labeling NCs on the package of healthier sausages is worthwhile to generate a better sensory experience and elicit higher WTP. Nevertheless, whether HCs information can generate some more positive impacts or more negative halo effects on sensory liking and WTP compared to NCs could be product specific.

There are several limitations to this study. One is that the sample population was not representative and was biased in age, education, and employment. Therefore, the conclusions are more applicable to a college consumer group. Another limitation is that participants might overestimate WTP values, as the CBC experimental approach is a stated preference method. Further studies are warranted to test NCs and HCs made on a wider variety of healthier meat prototypes among a representative sample population according to ISO recommendations (ISO 20784:2021). Non hypothetical experiments, such as auctions with real money transactions, could calibrate the implicit over-stated payment intentions. Eye-tracking studies could also provide data to further understand behavioral preference for food information and claims.

## Figures and Tables

**Figure 1 foods-11-03460-f001:**
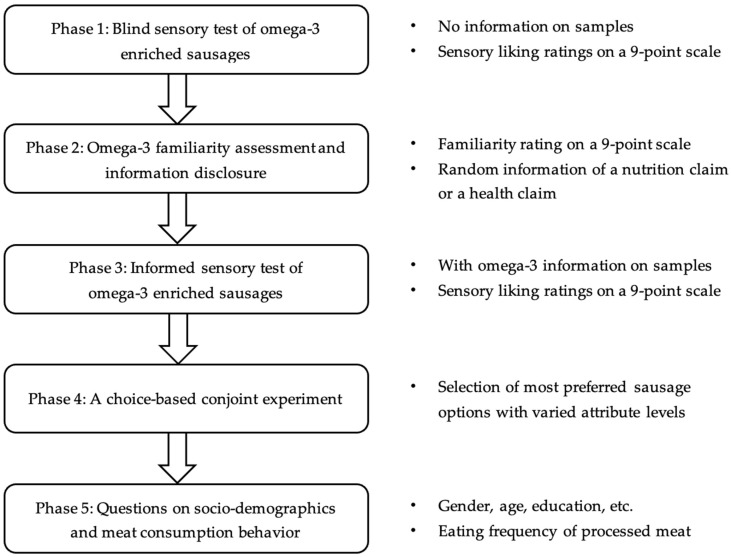
A diagram of the experimental procedure.

**Table 1 foods-11-03460-t001:** Attributes and levels of sausages used in CBC experiment.

Attributes	Levels	Notes
Price	Four Levels: EUR 2.60, EUR 2.80, EUR 3.00, EUR 3.20	Per pack price weighting 454 g
Meat Content	Four Levels: 60%, 70%, 80%, 90%	Pork meat percentage in sausages
Nutrition claim	Two Levels Source of Omega-3, No Omega-3	An eligible nutrition claim (as listed in the Annex to Regulation (EC) No. 1924/2006) of “source of omega-3 fatty acids”
	Two Levels: Reduced Fat, No Fat Reduction	An eligible reduced (name of nutrient) claim (as listed in the Annex to Regulation (EC) No. 1924/2006) meaning “reduced in saturated fatty acids”
	Two Levels: Reduced Salt, No Salt Reduction	An eligible reduced (name of nutrient) claim (as listed in the Annex to Regulation (EC) No. 1924/2006) meaning “reduced in sodium/salt”.

**Table 2 foods-11-03460-t002:** Characteristics of the total participants, N group and H group.

	Total	N ^1^ Group	H ^1^ Group	*p*-Value ^2^
N (persons)	326	163	163	
Gender (%)				0.578
Male	43	45	42	
Female	57	55	58	
Age class (%*)*				0.064
18–24	62	55	69	
25–34	26	33	20	
35–44	7	6	9	
45 and Over	5	6	3	
Education level (%)				0.401
Secondary or less	12	12	12	
College credit, no degree	20	19	21	
Bachelor	28	25	31	
Master or professional	31	36	26	
Doctoral or above	8	7	8	
Others	1	1	2	
Employment Status (%)				0.856
Student	74	74	74	
Employed Full-Time	20	19	20	
Employed Part-Time	5	5	5	
Not Employed	1	2	1	
Household income range (%)				0.700
EUR 15,000 and below	8	9	7	
EUR 15,001–EUR 40,000	28	29	27	
EUR 40,001–EUR 80,000	19	18	20	
EUR 80,001 and above	20	19	20	
Do not know or prefer not to answer ^3^	25	25	26	
Familiarity with omega-3 ratings (score) ^4^				0.197
Average rating	6	6	5	
Eating frequency of processed meat (e.g., sausages, nuggets, burge) (%)				0.443
15 or more times a week	4	5	2	
7–14 times a week	14	15	14	
4–6 times a week	23	23	22	
1–3 times a week	39	36	41e	
Less than once in a week	21	21	20	

Note: ^1^ N and H correspond to two claim information conditions, where N denotes an omega-3 nutrition claim information condition, and H denotes an omega-3 health claim information condition. ^2^
*p*-Value derived from independent two-sample t-tests of equal variances, with Ho suggesting a mean difference equaling zero. ^3^ About twenty-five percent of participants did not know or preferred not to answer the level of household income and were assigned an average income level for further analysis. ^4^ Omega-3 fatty acids familiarity was self-assessed using a Likert 9-point hedonic scale, where 1 = ‘‘extremely unfamiliar”, 5 = “neither unfamiliar nor familiar”, and 9 = “extremely familiar”.

**Table 3 foods-11-03460-t003:** Sensory liking means for omega-3-enriched sausages.

Modality		N ^1^ Group (*N* = 163)				H ^1^ Group (*N* = 163)			
		Blind ^2^	Informed ^2^	∆ ^3^	*p*-Value ^4^	Blind ^2^	Informed ^2^	∆ ^3^	*p*-Value ^4^
Appearance	Mean (S.D.)	5.24 ^a^ (1.77)	5.54 ^x^ (1.65)	−0.30	0.007 **	5.80 ^b^ (1.73)	5.96 ^y^ (1.61)	−0.15	0.217
Taste	Mean (S.D.)	5.33 (1.76)	5.42 (1.75)	−0.09	0.533	5.44 (1.84)	5.52 (1.82)	−0.07	0.610
Texture	Mean (S.D.)	5.01 (1.82)	5.36 (1.84)	−0.36	0.019 *	5.05 (2.00)	5.26 (1.86)	−0.21	0.106
Overall Liking	Mean (S.D.)	5.28 (1.63)	5.52 (1.72)	−0.25	0.075	5.41 (1.88)	5.55 (1.80)	−0.14	0.316

Note: * and ** denote significance at the 5%, and 1% levels, respectively. SD is short for standard deviation. A raw sausage sample and a cooked sausage sample were presented to participants in pairs on a food tray, where the raw was for evaluated for appearance and the cooked was evaluated for taste and texture. Sensory liking on raw sausage’s appearance and the cooked sausage’s taste, texture and overall liking were evaluated using a horizontal Likert 9-point scale, where 1 = “dislike extremely”, 2 = “dislike very much”, 3 = “dislike moderately”, 4 = “dislike slightly”, 5 = “neither like nor dislike”, 6 = “like slightly”, 7 = “like moderately”, 8 = “like very much”, and 9 = “like extremely”. ^1^ N and H corresponded to two information conditions regarding claims, where N denotes an omega-3 nutrition claim information condition, and H denotes an omega-3 health claim information condition. ^2^ Blind and informed corresponded to two information conditions regarding omega-3, where under the blind condition, participants tasted omega-3 sausages without being given any additional information, and under the informed condition, participants tasted omega-3 sausages, aware that they were omega-3 enriched. ^3^ ∆ represents the within-subject difference of the two sensory liking means before vs. after the omega-3 information disclosure. ^4^
*p*-Value derived from the paired t-tests with Ho suggesting a mean difference equaling zero. ^a,b^ Under the blind condition, blind omega-3 sausages with different letters in a row are significantly different (*p* ≤ 0.05, independent two-sample t-test with equal variances with Ho suggesting a mean difference equaling zero). ^x,y^ Under the informed condition, omega-3 sausages with different letters in a row, are significantly different (*p* ≤ 0.05, two-sample t-test with equal variances with Ho suggesting a mean difference equaling zero). S.D. is short for standard deviation.

**Table 4 foods-11-03460-t004:** Means and 95% confidence intervals of WTP estimation.

Variable	N ^1^ Group (*N* = 163)	H ^1^ Group (*N* = 163)
	Mean (S.D. ^2^)	Mean (S.D. ^2^)
Constant (neither)	−2.93 ** (2.28 **)	−3.41 ** (2.28 **)
Meat ^3^	3.24 ** (2.77 **)	3.07 ** (2.75 **)
Om3 claim	0.50 ** (0.47 **)	0.59 ** (0.59 **)
Rfat claim	0.59 ** (0.51 **)	0.43 ** (0.48 **)
Rsalt claim	0.39 ** (0.32 **)	0.43 **** (0.46 **)
Log-likelihood	−1332.31	−1289.12
Wald Chi-Square	627.06	545.31
AIC ^4^	2688.61	2602.24
No. of participants	163	163
No. of observations	5868	5868

Note: ** denotes significance at the 1% level. ^1^ N and H correspond to two information conditions regarding claims, where N denotes an omega-3 nutrition claim information condition, and H denotes an omega-3 health claim information condition. ^2^ S.D. is short for standard deviation. ^3^ When interpreting the coefficient of “meat content”, the coefficient should be multiplied by 0.1, indicating a 10% increase/decrease in meat content. ^4^ AIC is short for Akaike information criterion.

## Data Availability

The data presented in this study are available on request from the corresponding author.

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
