# Peer review of "The Effects of Nutrition and Health Claim Information on Consumers’ Sensory Preferences and Willingness to Pay"

_foods, 2022, doi:10.3390/foods11213460_

Round 1
Reviewer 1 Report
Nutrient and health claims have long been an important research topic, which has been examined from a number of perspectives. There have been many publications on sausages, mainly on price, salt reduction, sensory changes. Despite the recent exponential increase in the number of choice-based experimental food research studies, the novelty of this research is that it investigates consumers' sensory preferences in several steps, including how sensory preferences were altered by information on omega-3 fatty acid claims.
The title is too long, the abstract is too long, it should be shortened to one paragraph, so that the objective, material, method and results are included.
For search terms, it is advisable to include relevant terms that are not included in the title. It is worth correcting this.
The objective of the research should be more precisely defined in a paragraph. If the sensory tests were carried out in an ISO-standard sensory laboratory, it is worth referring to ISO 8589:2007 Sensory analysis - General guidance for the design of test rooms; ISO 13300-1:2006 Sensory analysis - General guidance for the staff of a sensory evaluation laboratory - Part 1: Staff responsibilities; ISO 13300-2:2006; Sensory analysis - General guidance for the staff of a sensory evaluation laboratory - Part 2: Recruitment and training of panel leaders.
A new diagram of the experiment should be prepared.
Five phases to be completed in one session:
(i) a blind sensory test of omega-3 enriched sausages; (More precise description is needed regarding: rotation of samples, taste neutralisers, whether 3-digit codes had a zero starting point, artificial daylight type illumination specification, why was category 1 "dislike extremely" in a structured 9-point scale, why not category 9).
(ii) asking participants' familiarity with omega-3 fatty acids and randomly informing participants of an omega-3 claim (either a nutrition claim or a health claim);
(iii) an informed sensory test of omega-3 enriched sausages;
(iv) a CBC experiment requiring participants to choose their preferred sausage options; (The determination of the properties and levels and the selection of participants were appropriate.)
(v) remaining questions regarding sociodemographic backgrounds and consumption habits of processed meat and dietary supplements.
A full-factorial set of attribute level combinations produced 128 product alternatives (4*4*2*2*2). A good choice of the Federov algorithm to reduce the number of alternative samples. The resulting 12 combinations of choices may be useful to upload into the supplementary material. The description of the econometric models and data analysis is sufficiently detailed and accurate, I found no errors.
The presentation of the results follows a logical structure, is generally good, and correctly compares own results with other international results. The conclusions are drawn correctly.
The socio-demographic characterisation of the participants is sufficiently detailed, it is fortunate that the sub-samples (N group, H group) were balanced, as this minimizes sampling bias. The authors also rightly mention this as a limitation in the conclusion, that the sample is not representative and biased in terms of age, education and employment.
Table 1 and sub-tables are adequate, perhaps it is worthwhile to fill in all preference categories.
In Table 3, the Mean and SD columns should be omitted, instead it would be useful to show only the mean SD value. For the P-values, I suggest to give values up to three decimal places, as they can be quite informative, especially for the sensory characteristics of appearance and texture.
The clarity of Table 4 would be improved if the decimal points were below each other. The *** should also be defined as it is included in the table. It would be advisable to omit the Mean and SD columns, instead only the mean SD value should be shown.
The conclusion is concise and to the point. A general methodological problem with preference surveys is that one of the major shortcomings is hypothetical bias, as individuals may behave inconsistently if they do not have real commitments to support their decisions, or respondents may not reveal their true preferences without real commitments, thus leading to overestimation of WTP values. At the end of the conclusion, more specific methodological directions (e.g. eye-tracking studies) might be more appropriate in the outlook. e.g. further studies might consider sensory and consumer product claims (ISO 20784:2021 Sensory analysis - Guidance on substantiation for sensory and consumer product claims).
In Appendix B (Table: An information summary of selected pork sausages available in Irish supermarkets), the last column is Pork content %.)
Reviewer 2 Report
Dear authors, After reading the manuscript "How European nutrition and health claims influence consumers’ sensory preferences and willingness-to-pay for healthier sausages ", I realized that the manuscript showed in some parts the scientific rigour wanted, but in other parts I have missed it.
The authors have presented critical evaluation only in some paragraphs.
The references are not exactly current, besides the objective could be shorter, please summarize the idea, you have 7 lines for the objective.
Thats why I have written some suggestions below in an attempt to improve the paper.
I request attention to the journal's guidelines for citing authors. Please, take a look at the Instructions for Authors.
References: References must be numbered in order of appearance in the text (including table captions and figure legends) and listed individually at the end of the manuscript. We recommend preparing the references with a bibliography software package, such as EndNote, ReferenceManager or Zotero to avoid typing mistakes and duplicated references.
L.1- Do you really need to include "European" in the title? I didn't realize this being addressed in your objectives, in the discussion, nor in the conclusion. Only in the Appendix, which confirms my perception that perhaps the importance given in the title is not adequate. Please, think about this.
L.42, L.45, L.48, L50 - The word "health" and its derivations became very repetitive. In some sentences it is 2x.
L.59-L.77 - Again, the word "health" and its derivations became very repetitive in the text
L.92-104 - Again, the word "health" and its derivations became very repetitive when reading.
L.112- 120- The same happens with the word omega-3, and it became very repetitive when reading
You mentioned health, but you didn't address the importance of omega for health, which is one of the focuses of the paper.
L.142- "asking participants’ familiarity with omega-3 fatty acids" - what exactly do you mean ?
L.148- I missed some information about the assessors. How many ? trained ? profile ? how were they invited/ approached ? What test was performed?
L.152- Were the participants the same in all the tests? It is not clear.
L.161- I know sensory analysis scales, but I think it is important to include the author, because not everyone who will read your paper will know about it.
L.170- author as well, please.
L.182- Shouldn't that part about the groups come first? I found it confusing.
L.329 -I think it is important that the "n" appears in the material and methods. The readers keep wondering until this point how many were the assessors.
L. 336- even though there was no statistically significant difference, do you think that the age group may have influenced the results found? could this be a bias?
L.353- It got confusing to understand in the table: "raw sausage", "cooked sausage". When it was mentioned in lines 159, 160 and 186, I created a different expectation, which I didn't find in table 3.
Appearance got misconfigured in the table 3.
Round 2
Reviewer 2 Report
Dear authors, there are still some adjustments in the configuration and authors' guide of the journal, but the next step is the diagramming/formatting staff will be advising you.